Journal of
open psychology data

# A Bilingual Dataset for Testing Web Probing in the US and India: The Example of Measures of Environmental Concern

DATA PAPER

INGRID ARTS 

RENS VAN DE SCHOOT

KATHARINA MEITINGER

*Author affiliations can be found in the back matter of this article

]u[ ubiquity press

## ABSTRACT

To assess the feasibility of web probing in India, we conducted a study with different probing questions and a survey experiment in India and the US on the topic of environmental concern. The survey was available in English in both countries and in Hindi in India. The data was collected from December 2022 until May 2023 using the crowdsourcing platform Amazon MTurk. This resulted in 512 respondents from the US and 740 from India (English: 387 respondents, Hindi: 353 respondents). The data provides crucial insights into the performance of web probing outside the Western context and the comparability of measures of environmental concern.

CORRESPONDING AUTHOR:
**Ingrid Arts**

Utrecht University, Netherlands

i.j.m.arts@uu.nl

KEYWORDS:
web probing; India; US; open-ended questions; environmental concern

TO CITE THIS ARTICLE:
Arts, I., van de Schoot, R., & Meitinger, K. (2024). A Bilingual Dataset for Testing Web Probing in the US and India: The Example of Measures of Environmental Concern. *Journal of Open Psychology Data,* 12: 13, pp. 1–12. DOI: https://doi.org/10.5334/jopd.113

# (1) BACKGROUND

Web surveys quickly reach respondents from many countries at relatively low costs (Lehdonvirta et al., 2020; Lenzner & Neuert, 2017; Mercer et al., 2017). However, the increase in cross-national surveys poses the challenge that respondents in different countries potentially understand some of the questions differently, i.e., the questions are not equivalent across countries. Comparing the responses to these questions would be like comparing apples and oranges (Horn & Mcardle, 1992). Before international survey data can be interpreted, it is necessary to assess whether the data is indeed comparable or whether differences across countries are mere methodological artifacts. To assess whether questions are valid and comparable across countries, it is possible to ask probing questions (Meitinger, 2017).

Probing is a technique in cognitive interviewing in which the interviewer asks follow-up questions to a closed target question to gain more insights into the understanding and reasoning of the interviewee before answering the closed question (Beatty & Willis, 2007; Meitinger & Behr, 2016; Priede et al., 2014). This technique has been adjusted for web surveys by Behr et al. (2014). They defined web probing as "the implementation of probing techniques from cognitive interviewing in web surveys with the goal to assess the validity of survey items" (Behr et al., 2014, p. 524).

Web probing usually distinguishes between three probe types: a category selection probe, a specific probe, and a comprehension probe. These probes are the same as in Behr et al. (2014).

1) A category selection probe asks why a respondent selected a certain answer category ("Why did you select 'very serious' when answering the previous question?")
2) A specific probe asks what detail(s) of a key term a respondent was thinking of when answering a target question ("Which problems relating to global warming or the greenhouse effect were you thinking of when answering the previous question?")
3) A comprehension probe asks how a respondent understands or defines a key term ("What do you consider to be 'poor air quality'?")

For a visualization of these probes, see Figure 1.

By asking these probes, it is possible to detect which survey questions potentially lack comparability. In addition, they reveal the reasons for missing comparability (Meitinger, 2017). For example, web probing can reveal that respondents apply the same reasoning but select different response categories for measures of environmental attitudes. For the statement "The government should reduce environmental pollution,

but it should not cost me any money," the same reasoning might lead to differences in the selection of an answer category on a response scale ranging from "completely agree" to "completely disagree." Respondent A might select the response category "agree" because she argues that "I already behave in an environmentally friendly way as much as possible, and I do not want to pay more." At the same time, respondent B might choose the response category "disagree" because he thinks that "I disagree because I do not behave in an environmentally friendly manner, so I should pay for that." Both respondents use similar arguments (paying for non-environmental friendly behavior) but select different responses. When taking only the answer to the closed question into account, the conclusion would be that there are differences between the answers when, in reality, there are none.

Web probing has already been successfully implemented in the US, Europe, and Mexico (Behr et al., 2019) and is recognized as a vital tool for the assessment of validity and comparability of survey items (Fowler & Willis, 2019; Geisen & Murphy, 2019; Silber et al., 2020; Singer & Couper, 2017). Extensive previous research on the optimal implementation of web probing exists (Behr et al., 2012; Edgar & Keating, 2016; Hadler, 2023; Lenzner & Neuert, 2017; Meitinger et al., 2018, 2021; Meitinger & Behr, 2016; Scanlon, 2019). For example, Meitinger & Kunz (2022) and Kunz & Meitinger (2022) assessed the optimal visual design of specific probes by experimentally manipulating the number of answer boxes in German web probing studies.

However, there is a research gap regarding the use of web probing in non-Western cultures. For example, does web probing perform equally well in a non-Western culture like the Indian one? Is it sufficient to translate only the established probe types or is it necessary to make additional adjustments (e.g., regarding wording, instructions, or visualization)? Do the probes collect sufficient insights to assess the comparability of questions and do they provide explanations for cases of missing comparability? To answer these questions, we conducted a web probing study in India and the US on the topic of environmental concern.

The topic of environmental concern was selected because this is a very pressing, universal topic: there are numerous environmental problems, and almost the entire planet has, at some point, been confronted with environmental change (Singh & Singh, 2017), either through personal experience or (social) media exposure. Despite the universality of the topic (Diekmann & Franzen, 2019), previous research found comparability issues regarding its measurement (Arts et al., 2021; Mayerl & Best, 2019), potentially due to differences in culture and exposure to environmental risk across countries. That makes this topic very useful for studying cultural influences on item understanding.

Arts et al. *Journal of Open Psychology Data* DOI: 10.5334/jopd.113

We selected India for this study because it has a very different culture than the American or European one (Krishnan & Poulose, 2016). Indian culture is considered to be collectivistic and hierarchical (Hofstede, 1980; Nishimura et al., 2008). Communication is more indirect (Junghare, 2015), respectful, and people-oriented. Indians are less direct to disagree, but use techniques like softening the disagreement (with softened negative statements, apologies) or delaying their answer (Al-Sallal & Ahmed, 2022). American culture is seen as individualistic, and the pursuit of autonomy and personal achievement is regarded important. Communication is direct and in line with a person's feelings (Hall, 1976, p. 76). This indicates that there is a difference between the way respondents in both cultures answer questions. Furthermore, India already has a survey culture (Lau et al., 2019), and its enormous population makes it a rich source of information.

The goal of the overarching project is to assess whether the method of web probing performs equally well in the US and India, and whether the different probe types show variations in performance across the countries. We asked three different probe types, and every probe type was asked at least two times: category selection probe (four times), comprehension probe (three times), and specific probes (two times). In addition, our study replicates the experimental study by Meitinger and Kunz (2022) in a cross-national context to assess whether visual manipulations in probe design can improve the performance of specific probes and whether the effect of the manipulation differs across countries. Moreover, the data described in the current paper allows the assessment of comparability of several measures of environmental attitudes from the World Value Survey. The current paper describes the process of survey design and data collection.

## (2) METHODS

### 2.1 STUDY DESIGN

For our web probing study, we conducted a cross-sectional survey in India and the US. Participants were recruited online via Amazon MTurk. Amazon MTurk is a crowdsourcing marketplace where members can, for a small incentive, complete virtual tasks, in our case a survey. Although the use of Amazon MTurk has been disputed (Aguinis et al., 2021), many challenges, such as high attrition rates or the use of bots, can be counteracted by offering fair compensation (Litman et al., 2014) or adding extra bot protection (Kennedy et al., 2020). Also, the use of more experienced MTurk users decreases anomalous responses (O'Brochta & Parikh, 2021).

**Figure 1** Target questions and the three different probe types.

**Figure 2** One answer box for specific probe (left) vs five list-style boxes (right).

We also conducted an experiment regarding the response categories of the specific probes. This experiment is a replication of the study of Meitinger and Kunz (2022). The experiment had a between-subjects design where respondents were randomly assigned to the experimental conditions at the beginning of the survey. Half of the respondents were randomly assigned to a condition with one answer box, and the other half of the respondents were assigned to a condition with five list-style answer boxes (see Figure 2).

## 2.2 TIME OF DATA COLLECTION

The field period for the survey was from December 2022 until May 2023. Every month, the survey was open to seven or eight different quota groups (age, gender, and country-language combination), see Table 1. Only after the survey had been closed to all survey groups, whether the quota was filled or not, was the survey reopened to new quota groups.

When respondents opened the survey, they had one hour to complete. Initially, the survey was open on Mturk for 21 days (or until the quota was reached). When the quota was not filled, the survey was extended for another ten days. After this, the survey was closed to that specific quota group. For the quota groups that had not been filled after 31 days, the survey was reposted in a second round in April/May 2023.

## 2.3 LOCATION OF DATA COLLECTION

The web probing study was conducted in the US and India via a web survey.

## 2.4 SAMPLING, SAMPLE, AND DATA COLLECTION

The intended sample size was 1,200 respondents, evenly distributed over three survey groups: US English (400), India with English language version (400), and India with Hindi language version (400). The participation in the survey was quota-based in all three survey versions. For every survey version, we aimed to get an equal distribution of male and female respondents and an equal distribution per age group of 18–30, 31–50, and 50+ years old.

### American English sample

Respondents in the US received the survey in English. In total, we received 512 responses from American respondents. This was rather peculiar since we only asked MTurk for 400 US respondents. This means that we received 112 extra responses. Even after data cleaning, we were left with 459 respondents. Although we were not sure where these extra respondents came from, we decided to keep them in the dataset since the answers they provided might contain useful information.

Of these 459 respondents, 203 were male, 207 were female, and one person chose the option "other". 48 respondents did not declare their gender. Ages ranged

between 21 and 72 years old (M = 36.7, SD = 12.2). In total, 371 respondents had completed higher education (bachelor or more), while 40 had not, and another 48 respondents did not answer this question or quit the survey before they reached this part of the survey.

### Indian English sample

Respondents from India could opt for the survey in English or Hindi. 387 Indian respondents started the English survey. After screening and exclusion (for details, see the section about data screening below), 373 English responses from India were included in the final dataset. Of these, 185 were male, 143 were female, and 45 did not provide this information. Ages ranged from 21 to 70 years old (M = 36.1, SD = 10.1); 312 respondents had completed higher education, 15 had not, and 46 respondents did not answer this question or quit the survey before they reached this part of the survey.

### Indian Hindi sample

The survey in Hindi yielded a total of 353 responses, of which 43 respondents were excluded from the dataset, with 310 respondents who answered the Hindi dataset. Of these, 153 respondents were male, 91 were female, 2 respondents selected "other", and 64 respondents did not answer this question. Ages ranged from 18 to 64 years old (M = 34.4, SD = 8.1); 228 respondents had completed higher education, 18 had not, and 43 skipped this question or quit the survey before they reached this part of the survey.

This means that the total sample size started with 1252 responses, and 1142 were included in the final dataset after data cleaning, which is 91.21% of the total responses. Data were excluded from the final dataset based on three criteria: the lack of a valid MTurk ID, break-off before the first question, and speeding. The obtained sample sizes can be found in Table 1.

Respondents who completed the survey received an incentive of $2.50. This coincides with the hourly minimum wage in the US (US government, n.d.). $2.50 is higher than the minimum wage in India, but how much is not exactly clear, as minimum wage differs per state, type of labor, and skill of the worker. We selected a comparatively high incentive for India because Litman et al. (2014) showed that compensation above minimum wage increased response quality for Indian Amazon Mturk workers.

## 2.5 MATERIALS/SURVEY INSTRUMENTS

The survey had 36 questions, of which 23 were closed. The survey contained nine probing questions. In addition, three open-ended questions asked for a numerical input (either a percentage or an age). Twenty-three of the 36 questions were about environmental concerns, two were postmaterialism measures (Inglehart, 1971), and eight were socio-demographic questions. The survey ended

| AGE (YEARS) | QUOTA M | ACHIEVED M MTURK | SR AGE-GENDER M | QUOTA F | ACHIEVED F MTURK | SR AGE-GENDER F |
|---|---|---|---|---|---|---|
| US | | | | | | |
| 18–25 | 34 | 34 | 33 | 34 | 34 | 75 |
| 26–30 | 34 | 34 | 44 | 34 | 34 | 26 |
| 31–35 | 34 | 34 | 46 | 34 | 34 | 27 |
| 36–45 | 34 | 34 | 30 | 34 | 34 | 29 |
| 46–55 | 32 | 32 | 32 | 32 | 32 | 18 |
| 55+ | 32 | 32 | 18 | 32 | 32 | 32 |
| NA | 48 | | | | | |
| India-English | | | | | | |
| 18–25 | 34 | 34 | 23 | 34 | 34 | 26 |
| 26–30 | 34 | 34 | 40 | 34 | 34 | 29 |
| 31–35 | 34 | 34 | 41 | 34 | 34 | 27 |
| 36–45 | 34 | 34 | 46 | 34 | 34 | 44 |
| 46–55 | 32 | 26 | 22 | 32 | 5 | 10 |
| 55+ | 32 | 1 | 13 | 32 | 10 | 7 |
| NA | 45 | | | | | |
| India-Hindi | | | | | | |
| 18–25 | 34 | 34 | 12 | 34 | 13 | 13 |
| 26–30 | 34 | 34 | 37 | 34 | 22 | 24 |
| 31–35 | 34 | 34 | 49 | 34 | 17 | 17 |
| 36–45 | 34 | 33 | 39 | 34 | 15 | 28 |
| 46–55 | 32 | 7 | 13 | 32 | 6 | 7 |
| 55+ | 32 | 2 | 3 | 32 | 1 | 2 |
| NA | 64 | | | | | |

**Table 1** Samples size: Quota, MTurk output, and self-reported age and gender.

M: male, F: female, Quota: desired number of responses, Real MTurk: gender-age based sample size from MTurk settings, SR age-gender: self-reported age and gender.

with three questions on survey evaluation. For a full overview of the questions and their order, see Appendix A, or the codebook on https://dans.knaw.nl/nl/social-sciences-and-humanities/.

We measured three constructs about environmental concern: The Willingness to Pay to save the environment (WTP), the seriousness of local environmental problems, and the seriousness of global environmental problems. These three constructs were also assessed by Knight and Messner (2012). Each construct was measured by three indicators, so-called target questions. The target questions had a four-point answer scale: 'strongly agree', 'agree', 'disagree', and 'strongly disagree' for the construct WTP, and 'very serious', 'somewhat serious', 'not very serious', 'not serious at all' for the constructs about local and global seriousness of environmental problems. A 'don't know' option was available, but this option was visually set apart from the other response

categories (see Figure 1). A probing question followed every target question in a construct. The construct WTP had two additional questions, asking respondents which percentage of their income they are willing to give/which percentage of tax increase they are willing to accept if they can be sure the money is spent to protect the environment. The other two constructs each comprised six questions, with one construct asking about the perceived seriousness of environmental problems in the local area and one construct about the perceived seriousness in the world as a whole.

Throughout the survey, we used a page-per-page design. After answering the target question, respondents received the probe on a separate screen. Answer boxes for the probes were designed as advised by Meitinger et al. (2018). To reduce response burden, the target question was always repeated after the probing question, see Figure 1.

### Between subjects design

The between-subjects design randomly assigned respondents to a control or experimental version of the survey. For the control version, probes were designed as proposed by Behr et al. (2014), with smaller answer boxes for the specific probes and larger answer boxes for the comprehension and category selection probes. For the experimental version, respondents were asked to write down the answers to the specific probes in five small answer boxes (list-style presentation). Meitinger and Kunz (2022) found for Germany that this design increased the number of themes and succinctness of the responses given.

### *Questions*
#### *Closed questions*

To assess their comparability, the original question wording from the World Value Survey 5 source questionnaire and the translations was used for the closed-ended questions about environmental concern, as well as the questions about postmaterialism (Inglehart et al., 2018).

#### *Qualitative data*

In total, there were ten open-ended questions: nine probing questions and one question at the end of the survey. This last question asked people if they had any further comments about this survey.

#### *Background questions*

To obtain more background information, we asked several socio-demographic questions. We asked people about their gender, age, education, religion, the place they lived (a big city to a rural area), and their income relative to other families/households in their country. For exact question wording, see Appendix A.

The socio-demographic questions about gender, age, and education had a hard reminder, meaning that respondents who skipped this question could not continue until they answered this question. Several respondents skipped (one of these) questions. Therefore, it seems that this hard reminder did not always function correctly.

#### *Movement through questionnaire*

Respondents could move back and forth in the survey by selecting the left or right-pointing arrows. However, when a question was not answered, respondents received a soft reminder indicating that they skipped a question, followed by the question of whether they wanted to proceed without answering the question or go back and answer the question (Figure 3).

### 2.6 QUALITY CONTROL

### Material Production (Translation Procedure)

As mentioned above, the majority of the target questions were replicated from the original questions from the

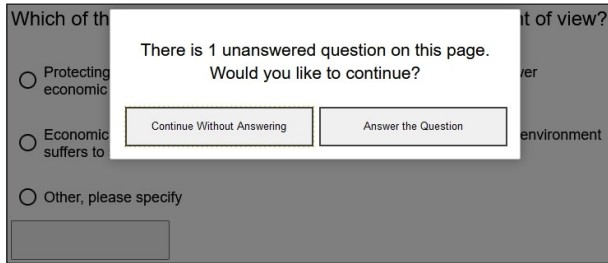

**Figure 3** Soft Reminder.

WVS 5 questionnaires. We used the English source questionnaire and the WVS translations for India (English and Hindi) and adapted them to the web mode.

Socio-demographic questions were updated by comparing questions and response categories to more recent questionnaires (e.g., WVS wave 6, ISSP Background Variable Questionnaire 2018). Questions and response categories were adapted to culture where necessary. We used the probe types as proposed by Behr et al. (2014).

The US version of the survey served as the source questionnaire. For the English Indian version, several background questions or response categories, such as education, religion, or the state respondents live in, needed to be adapted to fit the Indian context. We used background variables from the ISSP (Davern et al., 2021; Edlund et al., 2015). For the background education variable, we used the official ISCED classification (UNESCO Institute for Statistics, 2015). In addition, two Indian project members provided cultural insights and checked for inconsistencies and impoliteness in formulations.

The English version of the Indian survey was then translated to Hindi following the TRAPD approach (Harkness, 2003). This process consists of a cycle of translating, reviewing, adjusting, pretesting, and documentation (Figure 4, Harkness et al., 2010). We conducted the following steps:

1) Translation: Two different translation companies, Blauvelt Vertalingen, and SIGV Corporation, and a native speaker independently translated the English version of the Indian survey to Hindi.
2) Reviewing: We discussed the final version of the Hindi translation with a native speaker in an extensive review meeting. Where needed, a second native speaker was consulted in this process.[1]
3) Adjudication: The translation was compared with the source questionnaire, and where necessary, some last adaptations were made in consultation with the Hindi speaker.
4) Pretesting. Due to time and budgetary issues, there was no cognitive pretesting phase, as recommended by Harkness (2003). We performed quantitative pretests of all three language versions to assess any malfunctioning (both substantive and technical)

of the questionnaire. Any inconsistencies or malfunctioning of the questionnaire were corrected and the survey was tested again.

**5)** Documentation: The whole process was documented in an Excel document, containing all three independent translations, the final translation, as well comments regarding the translation procedure.[2]

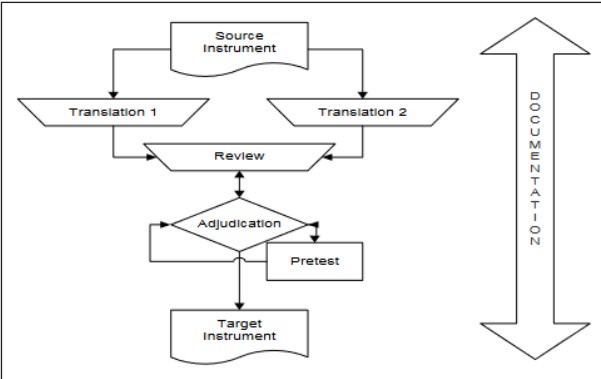

**Figure 4** TRAPD procedure (Harkness et al., 2010).

## Data quality and bot protection

The final versions of the questionnaires were implemented in Qualtrics™. To ensure that only the targeted audience (respondents from India and the US) could take the survey, only users with an IP address within these countries were allowed to take the survey. Respondents from outside these countries, as well as users of Proxy or Virtual Private Server (VPS), were blocked from the survey. For this purpose, we implemented IP HUB in our survey, as demonstrated by Kennedy et al. (2020).

To prevent bots from taking the survey, both reCAPTCHA v2 and v3 were used. At the beginning of the survey, respondents were asked to verify that they were not a robot by selecting certain parts of an image (reCAPTCHA v2). Throughout the survey, reCAPTCHA v3 was running in the background. This reCAPTCHA returned a number varying from 0 to 1.0, with a number below 0.3 being most likely bots and above 0.7 being most likely human. However, a low reCAPTCHA score can also be due to the used browser or the fact that JavaScript was disabled or not up to date. Therefore, answers given by respondents with no reCAPTCHA score or a reCAPTCHA score below 0.3 were manually checked. In that case, both the click count (number of times respondents click on a page) and the answers to the probing questions were compared to respondents with high reCAPTCHA scores.

As an additional quality measure, only MTurk workers who had a 90% or higher success rate in completing previous tasks on MTurk could participate in the survey since O'Brochta & Parikh (2021) found that this procedure can increase response quality.

Respondents were blocked from taking the survey twice by turning on the 'prevent multiple submissions' option in Qualtrics.

## Paradata

Paradata, such as clicking behavior, response duration, or browser type, are variables that contain additional information about the survey. They can provide information regarding missing data or break-off (Kunz & Hadler, 2020). Paradata can also help to identify bots. Storozuk et al. (2020) stated that looking at e.g., time of survey completion and speed of survey completion can help to determine bots.

For the overall survey, we recorded:

- Date: start date, finished date, and recorded date;
- Percentage of survey finished;
- Survey duration in seconds;
- Browser type and version;
- Operating system;
- Resolution.

In addition to this, we recorded, for every question (every survey page):
- First Click: How many total seconds the page was visible before the respondent clicks the first time;
- Last Click: How many total seconds the page was visible before the respondent clicks the last time (not including clicking the next button);
- Page Submit: How many total seconds pass before the respondent clicks the next button (i.e., the total amount of time the respondent spends on the page);
- Click count: How many total times the respondent clicks on the page.

## Technical functioning of the survey

Although neither the system nor any respondents reported any malfunctioning, it seems that the systems (MTurk and Qualtrics) did not always function as expected. First, although we requested specific gender and age quotas from MTurk, respondents who might not belong to this quota also filled in the questionnaire. We tested this by also asking respondents about their age and gender at the end of the survey. Aguinis et al. (2021) showed that more than 20% of MTurk respondents misrepresent on age questions, and 6% misrepresent on gender questions to fit the survey criteria. Questions about gender and age were asked at the end of the survey when specific survey requirements were more likely to have been forgotten, and respondents were more likely to answer truthfully. We, therefore, decided to use self-reported age and gender when analyzing these variables. Second, although the system was set to force respondents to answer the gender, age, and education questions, some of the respondents did skip (one or two of) these questions. Third,

                                                                                 

it appeared not all paradata were recorded: for part of the data browser type and – version, operation system, and resolution have not been recorded.

## Data screening

The data cleaning process comprised several steps:

1) All responses without a (valid) MTurk ID were dropped from the final data set: 51 respondents were dropped at this step.
2) Respondents who broke off the survey directly after the welcome or instruction screens were excluded from the final data set: 17 respondents were dropped at this step.
3) Although the effect of speeders (respondents going through the questionnaire answers without properly reading/answering the questions) can be considered marginal (Greszki et al., 2015), we did check the total time that was spent on the survey. When the total survey response time was below 60% of the median response time (Rossmann, 2010), an additional check was done on the data provided.

This means that, for respondents with very low survey response time (568.2 seconds for the US, 729.6 seconds for India in English, and 637.2 seconds for India in Hindi), we verified that the system indicated they had completed 90% or more of the survey. That data was manually checked to see if the open-ended questions had been answered in a poor way (e.g., all NA's, all keystrokes). Based on response time and questionnaire completion, 184 respondents could be flagged as speeders (US: 92, India_English: 60, India_Hindi: 42). However, the responses given by these respondents appeared to be quite useful, so no data was removed in this step. The sample sizes by country_language for each step in the cleaning process are shown in Table 2.

| | TARGET | INITIAL | MTurk ID | STARTED AT WELCOME SCREEN | FINAL |
|---|---|---|---|---|---|
| US | 400 | 512 | 480 | 464 | 459 |
| IE | 400 | 387 | 381 | 377 | 373 |
| IH | 400 | 353 | 340 | 334 | 310 |

**Table 2** Sample Size by Exclusion Criteria.

The difference between respondents who started at the welcome screen and those who are include in the final dataset is explained by respondents breaking of after the welcome screen, but before the instruction screen or introduction screen.

## 2.7 DATA ANONYMIZATION AND ETHICAL ISSUES

We did not process any personal data.

At the beginning of the survey, respondents saw a screen explaining that the survey was anonymous and that data would be processed according to the GDPR. Respondents could not continue unless they selected the next button (which was an arrow pointing to the right), thus giving informed consent. IP addresses were not associated with data, nor were they exported. Any data that might relate to IP addresses has been removed from the raw datasets. This study was approved by the Ethics Review Board of the Faculty of Social and Behavioral Sciences of Utrecht University, under number 22-017 on 13-05-2022.

## 2.8 EXISTING USE OF DATA

Until the current date, there are no publications that have originated from this dataset.

# (3) DATASET DESCRIPTION AND ACCESS

## 3.1 REPOSITORY LOCATION

The datasets are available on the Data Archiving and Networked Services (DANS) website: https://dans.knaw.nl/nl/.

## 3.2 OBJECT/FILE NAME

The following files are available:
- codebook.docx
- US_anonymized_raw.sav
- India_E_anonymized_raw.sav
- India_H_anonymized_raw.sav
- E_trans_textq.xlsx
- script_cleaning_data_ind_usa.rmd
- cleaned_data_India_US.xlsx
- cleaned_data_India_US.sav

## 3.3 DATA TYPE

Primary data
- Codebook: Word
- Raw data: available in SPSS, three datasets each
- Translations Hindi: Excel
- Code: RMarkdown

Final data

The Final dataset contains the data with all three country-language versions, and qualitative and quantitative data combined: both Excel and SPSS. The final dataset is saved in both Excel and SPSS file format. The file contains 234 variables and 1142 respondents.

## 3.4 FORMAT NAMES AND VERSIONS

- Excel: can be opened with several software packages, such as Microsoft Office, Google Sheets, LibreOffice Calc, Apache OpenOffice. Some of these packages are open source, and freely available.

- SPSS: can be opened with IBM SPSS statistics (license required). An SPSS sav file can also be read into R and then saved as a different file format.
- R Markdown: can be opened in R Studio, which is open source.
- Word: can be opened with several software packages, such as Microsoft Office, Google Docs or Libre Office.

### 3.5 LANGUAGE

The main language of the dataset is American English. The raw qualitative data are also in Indian English and Hindi. The language of the final dataset is only English (American and Indian).

### 3.6 LICENSE

CC-BY-NC-4.0

### 3.7 LIMITS TO SHARING

Embargo until September 2025

### 3.8 PUBLICATION DATE

26/04/2024

### 3.9 FAIR DATA/CODEBOOK

The codebook is available as a Word file on https://dans.knaw.nl/nl/social-sciences-and-humanities/.

Findability
DOI: https://doi.org/10.17026/SS/IFM3QS

Accessibility
Open access, embargo until March 2026

Interoperability
Excel is an open file format, but the use of an RMarkdown file requires R and RStudio. Though not everybody might know how to use R and RStudio, it is free and open-access software. The RMarkdown file is written in such a way that beginners can also run the file and generate output. While SPSS files can only be opened when an IBM license, an SPSS file can be loaded into R and RStudio and can be stored as another file format. We loaded the raw SPSS datasets into R, combined them with the Excel translation set, and after cleaning wrote them to the cleaned dataset, both available in Excel and SPSS format.

Codebook for interpreting raw data
The codebook for this data can be found on the DANS website.

## (4) REUSE POTENTIAL

The dataset we created can serve as a first stepping-stone for assessing the feasibility of web probing in India and the US. It can provide insights into the different thought processes regarding details of a key term (specific probe, number of details, originality), motivating answers (category selection probe), or understandings of a key term (specific probe).

On a more substantive level, this dataset can provide insights into the (lack of) comparability of several constructs measuring environmental concern, namely the willingness to pay to save the environment, the perceived seriousness of local environmental problems, and the perceived seriousness of global environmental problems. At the same time, the web probes provide the reasons for missing comparability. These insights are crucial information for improving existing measures for future waves of data collection.

## NOTES

1 In some cases, a literal translation was possible, but would not result in what would be considered usable Hindi. In that case, the translation adapted to Hindi. For instance, where English uses "The government should reduce environmental pollution, but it should not cost me any money" Hindi uses "The government should try to reduce environmental pollution, but I am not willing to pay any money". This is as close to the original WVS wave 5 translation as possible.

2 File available on request.

## ADDITIONAL FILE

The additional file for this article can be found as follows:

- **Appendix A.** Exact wording of questions and text elements in the questionnaire. DOI: https://doi.org/10.5334/jopd.113.s1

## ACKNOWLEDGEMENTS

The authors wish to thank Rishabh Suresh and Vidhi Ramnarain for their translation efforts and cultural insights.

## COMPETING INTERESTS

The authors have no competing interests to declare.

## AUTHOR AFFILIATIONS

**Ingrid Arts** orcid.org/0000-0003-2378-3354
Utrecht University, NL
**Rens van de Schoot** orcid.org/0000-0001-7736-2091
Utrecht University, NL
**Katharina Meitinger** orcid.org/0000-0001-8160-556X
Utrecht University, NL

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

## PEER REVIEW COMMENTS

*Journal of Open Psychology Data* has blind peer review, which is unblinded upon article acceptance. The editorial history of this article can be downloaded here:

- **PR File 1.** Peer Review History. DOI: https://doi.org/10.5334/jopd.113.pr1

Arts et al. *Journal of Open Psychology Data* DOI: 10.5334/jopd.113

**TO CITE THIS ARTICLE:**
Arts, I., van de Schoot, R., & Meitinger, K. (2024). A Bilingual Dataset for Testing Web Probing in the US and India: The Example of Measures of Environmental Concern. *Journal of Open Psychology Data,* 12: 13, pp. 1–12. DOI: https://doi.org/10.5334/jopd.113

**Submitted:** 01 May 2024    **Accepted:** 21 November 2024    **Published:** 06 December 2024

