## [Peer Review History. · Journal of Open Psychology Data]

Peer Review Report for "A bilingual dataset for testing web probing in the US and India: The example of measures of environmental concern"

Dear Ingrid Arts, Rens Schoot van de, Katharina Meitinger,

After review, we have reached a decision regarding your submission to Journal of Open Psychology Data, "A bilingual dataset for testing web probing in the US and India: The example of measures of environmental concern". Our decision is to request revisions of the manuscript prior to acceptance for publication.

The full review information is included at the bottom of this email. In summary, the work is perceived to be valuable but there are areas where a little more detail would help improve the clarity of the work. Primarily however, the data is currently inaccessible so does not meet the requirements of the JOPD. Please reconsider how you share your data and we will be happy to reconsider the submitted work,

Instructions for how to resubmit your article online are pasted below. Please ensure that your revised files adhere to our author guidelines, and that the files are fully proofed prior to upload. Please also include a revised version of your article with 'tracked changes', adding comments where appropriate, to indicate the revisions made, in addition to a brief document outlining how you have responded to the reviewers' requests.

If you have trouble processing the revisions, our Help Center (<https://help.u-community.io>) or downloadable PDF (<https://bit.ly/Author-Guide-OJS-3>) may be able to help. If not, please get in touch and we'll be happy to help.

Please also ensure that all copyright permissions have been attained for any figures/tables you have included.

Please could you have the revisions submitted with six weeks. If you cannot make this deadline, please let us know as early as possible.

Kind regards,

Dr Thomas Rhys Evans

Reviewer H:
Recommendation: Revisions Required

Comments to the author(s)

The manuscript “A bilingual dataset for testing web probing in the US and India: The example of measures of environmental concern” tries to test web probing procedures in order to determine their usefulness in assessing comparability of assessments in cross-cultural studies. This is a very important topic, because a lack of comparability can compromise validity and fairness in testing, and also because of the rise of web probing methods in modern research.

The manuscript is very clear and thorough in its “method” section, providing plenty of detail needed to understand how the dataset was developed, and how data was collected. Several approaches were followed to ensure data quality, in both data collection and data cleaning processes, which is very positive. The dataset is clearly described. The reuse potential is mentioned, but there is still a need to provide concrete suggestions for reuse of data. Most of the dataset files are not accessible yet due to being embargoed. The codebook is the only available file. It is needed to provide an open version for the rest of the dataset files, in order for the reviewers to properly see them.

Reviewer N:
Recommendation: See Comments

Comments to the author(s)

The data paper is descriptive. It includes sufficient details for the readers about the paper. The background of the study has sufficient details, as evidenced by the literature reviews. Kindly provide a citation or a rationale for the statement: Because the topic is so universal, differences in answers are more likely to be influenced by culture than by, e.g., a lack of knowledge.

Data papers are in short formats, but some of the important details leading to the rationale of the study, especially concerning Indian culture, are to be added. Why is it a different culture? What makes the difference stand out? Why is there a need for comparison?

Sufficient details about the survey, questions, response options available, and response interpretation are given.

The codebook is the only file openly available on the repository. It would be difficult to understand about the data and provide feedback without access to it.